# Estimating Depressive Symptom Class from Voice

**DOI:** 10.3390/ijerph20053965

**Published:** 2023-02-23

**Authors:** Takeshi Takano, Daisuke Mizuguchi, Yasuhiro Omiya, Masakazu Higuchi, Mitsuteru Nakamura, Shuji Shinohara, Shunji Mitsuyoshi, Taku Saito, Aihide Yoshino, Hiroyuki Toda, Shinichi Tokuno

**Affiliations:** 1Department of Bioengineering, Graduate School of Engineering, The University of Tokyo, Tokyo 113-8656, Japan; 2PST Inc., Yokohama 231-0023, Japan; 3School of Science and Engineering, Tokyo Denki University, Saitama 350-0394, Japan; 4Department of Psychiatry, National Defense Medical College, Tokorozawa 359-8513, Japan; 5Graduate School of Health Innovation, Kanagawa University of Human Services, Kawasaki 210-0821, Japan

**Keywords:** voice analysis, major depressive disorder, decision tree

## Abstract

Voice-based depression detection methods have been studied worldwide as an objective and easy method to detect depression. Conventional studies estimate the presence or severity of depression. However, an estimation of symptoms is a necessary technique not only to treat depression, but also to relieve patients’ distress. Hence, we studied a method for clustering symptoms from HAM-D scores of depressed patients and by estimating patients in different symptom groups based on acoustic features of their speech. We could separate different symptom groups with an accuracy of 79%. The results suggest that voice from speech can estimate the symptoms associated with depression.

## 1. Introduction

According to the WHO fact sheet [1], 280 million people worldwide were affected by depression in 2019. Furthermore, nearly 700,000 people die each year from depression-related suicide. This is the fourth leading cause of death among people aged 15–29 years, and the early detection and intervention of depression is a social issue.

When diagnosing depression, physicians use criteria such as DSM-5 and questionnaires or interviews to survey the severity of the condition. These can be an interview, such as the Hamilton Rating Scale for Depression (HAM-D) [2] or the Montgomery–Åsberg Depression Rating Scale (MADRS) [3], or a self-administered questionnaire, such as the Beck Depression Inventory (BDI) [4] or the Patient Health Questionnaire-9 (PHQ-9) [5]. These methods are widely used because they do not require special equipment and are not invasive. However, there is still a possibility that the subjectivity of physicians and patients may influence the diagnosis. For example, when taking these tests for reasons related to a patient’s job, he/she may provide incorrect information about their medical condition so as not to disadvantage them in their salary or career.

Therefore, methods using biomarkers such as saliva and blood have been studied as objective indicators. However, they have the disadvantages of being invasive, burdening subjects, and are costly due to the need for special measuring equipment and reagents.

As a method that does not have these disadvantages, research on voice biomarkers to infer depression from voices is being conducted using prosody analysis, frequency analysis, and glottal flow model analysis.

Prosodic analysis is focused on the fundamental frequency and speech velocity, which determine the pitch of the voice. Nilsonne and Sundberg et al. reported that emotional states could be discriminated by focusing on the fundamental frequency, without the need for a psychologist [6], and the computed fundamental frequency showed significant differences between depressed and improved states in depressed patients [7]. Other acoustic features such as jitter, shimmer, and f0 variability tended to increase with depression severity and psychomotor retardation, which affected motor control precision and laryngeal muscle tension [8]. In addition, the harmonic-to-noise ratio (HNR), a speech feature used to evaluate hoarseness, has been reported to be higher in depressed patients [9].

Hargreaves et al. compared the spectral analysis of the speech sounds of depressed patients with a mood index determined by two physicians [10]. Tolkmitt et al. reported that spectral energy below 500 Hz shifted from the 500 Hz to 1000 Hz band as the severity of depression increased [11].

The relationship between glottal flow and depressive symptoms has also been studied. After Alku published the Interactive Adaptive Inverse Filtering (IAIF) method [12], it was possible to estimate glottal flow. Using IAIF, the relationship between the Normalized Amplitude Quotient (NAQ) and Quasi-Open Quotient (QOQ) of glottal flow and depression has also been studied. Scherer et al. showed that there were significant differences in NAQ, QOQ, standard deviation (std) NAQ, and std QOQ between the no-depression and depression groups [13].

Moreover, machine learning has recently become more common as computational power has increased. By using machine learning methods, the accuracy when detecting depression and predicting severity scores has been compared. Williamson achieved a correlation coefficient of 0.56 for BDI scores with an analysis centered on formant frequencies, resulting in 0.53 with an analysis centered on Mel-cepstrum frequency differences and 0.7 using both [14].

The methods studied so far aim to improve the accuracy of detection for depression, using the presence of depression by physician diagnosis or questionnaire as a reference, and correlating it with significant differences and scores. On the other hand, the symptoms of depression are various, and the objective indices described above are judged comprehensively based on the severity of these symptoms. Antidepressants are the basic medication for depression, but they may be combined with anxiolytics or sleeping pills to alleviate patient’s distress. Estimating the presence or absence, as well as the severity of depression is not enough to establish such a treatment plan; estimating symptoms is also essential.

Therefore, we investigated the possibility of an algorithm that can estimate the presence or absence of depression and its symptoms, and support it as a treatment plan in a clinical setting by using these acoustic parameters known to be related to depression.

## 2. Methods

### 2.1. Ethical Considerations

This study was conducted with the approval of the Ethical Review Board of the Graduate School of Medicine, the University of Tokyo, under review no. 11572, “Research on the Development of Technology for Analyzing Pathological Conditions Using Speech”.

### 2.2. Data Acquisition

#### 2.2.1. Subjects

We targeted stress-related disorders in this study, namely: depression and bipolar disorder. To conduct the study, we collected audio recordings from depressed patients labeled with depressive symptoms.

We recruited subjects from depressed patients at the National Defense Medical College Hospital in Japan between 17 February 2017 and 5 July 2018. We recruited outpatients diagnosed with depression and undergoing treatment to participate in the study after they gave informed consent and we obtained research consent. We excluded those under 16 years of age or those unable to declare their intent to participate in the study.

#### 2.2.2. Recording

We recorded the subjects’ speech in the examination room in the hospital, using a Roland R-26 high-precision recorder with an Olympus ME-52W pin-type microphone attached to the subjects’ chests. The microphone was mounted approximately 15 cm away from the mouth. The audio format used for the analysis was linear pulse code modulation (PCM) with a sample rate of 11.025 kHz and a quantization bit of 16 bits, 1 channel.

We conducted the recording in the form of a subject reading out the fixed phrases twice. The recording duration was approximately 1 min. Table 1 shows the fixed phrases and their intentions.

#### 2.2.3. GRID-HAM-D

The GRID-HAM-D_17_, a 17-question version of the GRID-HAM-D [15], a form of the HAM-D, was used to label depressive symptoms. The episodic content of the GRID-HAM-D_17_ is shown in Table 2. The tendency of the scores was used to label the subjects’ symptoms.

When we use the term HAM-D hereafter, we refer to the GRID-HAM-D_17_.

### 2.3. Classification of Subjects According to Depressive Symptoms

The subjects in this study were interviewed and scored on the HAM-D to quantify symptoms. However, treating the question scores as independent indicators of symptoms was impossible because of the HAM-D questions’ correlations. For example, there was a correlation coefficient of about 0.7 in the data set between the first question, “depressed mood”, and the seventh question, “work and activity”. At this point, if each question was treated independently, we could not distinguish which item affects the acoustic features.

Therefore, we classified subjects based on the characteristics of the groups clustered by HAM-D instead of using the HAM-D scores directly, referring to a previous study [16]. We performed clustering using the following procedure.

First, we excluded those subjects with a total HAM-D score of 0 whose symptoms could not be estimated. This is because this study aimed to estimate symptoms, not severity, and we did not use this group for this study. Next, subjects with a HAM-D score of 4 or higher were selected and clustered into two groups using the K-means method. Finally, we allocated subjects with scores between 1 and 3 to the nearest of the groups that we created. We used the inverse of the correlation coefficient to determine the distance between subjects.

This study was not blinded because the operator who conducted clustering to identify symptom groups and the operator who estimated the symptom groups were the same person.

### 2.4. Prediction of Subject Classes

#### 2.4.1. Acoustic Features

Based on prior studies about depression and speech [7,8,11,13,14], we selected the acoustic features used to train the prediction model in Table 3. NAQ and QOQ are features that can quantify the state of the vocal cord from the voice. Fast fourier transform (FFT) band power is a feature that shows the power of the voice in each of three levels, namely: 0–500 Hz, 500–1000 Hz, and 1000–4000 Hz. FFT spectral centroid is a feature that represents the center of speech power, and FFT spectral flux is a feature that represents the transition of speech power from the high-frequency domain to the low-frequency domain, or vice versa. Formant covers the range from the first formant (Formant 1st) to the fifth formant (Formant 5th), and is a feature indicating the state of the vocal tract. Mel-frequency cepstral coefficients (MFCC) represent the spectral content of audio or speech signal, and are widely used in speech and audio processing applications. The energy root mean square (RMS) represents the loudness of the voice. Fundamental frequency, HNR, jitter, and shimmer describe the vocal cord state. Fundamental frequency is the frequency of the vocal cord’s wave, and the others indicate the stability of the waveforms.

For the calculation of the acoustic features, DisVoice [17] was used for the glottal flow features, openSMILE [18] for the frequency features, and Praat [19] for the prosody features. To facilitate machine learning, we used the Python library implementations praat-parselmouth ver. 0.4.0, DisVoice, and opensmile-python ver. 2.2. As openSMILE and DisVoice can output frame by frame, statistical processing of the standard deviation was introduced in addition to the mean value.

#### 2.4.2. Dataset

As a result of the voice collection, we obtained the phrases shown in Table 1. However, because each phrase had a different character for the same acoustic feature, we treated the acoustic features in different phrases as independent variables when predicting the model. As a result, we created 1032 independent variables per subject.

We used decision tree learning to predict the subjects’ symptom classes. We utilized the Python library LightGBM ver. 3.2.1 [20] for decision tree learning. To verify the model’s generalization performance, we conducted a five-fold cross-validation.

#### 2.4.3. Confusion Matrix

We determined the model’s accuracy as follows: using the test data from each of the five-fold cross validations, we determined the cutoff from the receiver operating characteristic (ROC) curves using the Youden Index and created a confusion matrix. We added the five results together to create an overall confusion matrix. From the confusion matrix, we calculated the sensitivity, specificity, and accuracy.

## 3. Results

### 3.1. Composition of Subjects

In the recruitment of subjects, we were able to gather 110 subjects. The average age of the male subjects was 54.8 years, and the average age of the female subjects was 59.9 years, indicating no significant difference in the distribution of male and female subjects. Table 4 shows the age and sex composition of the male and female subjects. All of the subjects were Japanese.

### 3.2. Results of Clustering Depressive Groups

Table 5 shows the groups created by clustering using HAM-D scores and their number. Table 6 shows the average HAM-D scores and the medication dose of the group.

Figure 1 illustrates the mean scores for each question of the HAM-D for group 1 and group 2. Asterisks in the figure indicate the pair with significant differences according to the Wilcoxon rank sum test. As the significance tests for the two groups were conducted 18 times, including those mentioned above for the HAM-D total scores, we corrected the significance levels based on the Bonferroni method. Consequently, we set the significance level at *p* < 2.8 × 10^−3^, where * is the commonly used 0.05 divided by 18. In the same way, ** and *** denote significance levels of *p* < 5.5 × 10^−4^ and *p* < 2.8 × 10^−4^, respectively. The significance level of the total score of HAM-D was *p* = 0.0078, which did not meet the significance level of *p* < 2.8 × 10^−3^. Therefore, we could not find a difference in the severity of depression between Group 1 and Group 2.

### 3.3. Result of Predicting the Depressive Symptom Class

Figure 2 illustrates the distribution of LightGBM outputs. The result includes all cross-validation test data.

Table 7 shows the confusion matrix of cross-validation results. We calculated the Youden Index of the training data for each cross-validation as the cutoff value. We also calculated a sensitivity of 83%, a specificity of 76%, and an accuracy of 79% from the confusion matrix.

Table 8 lists the features included in the prediction model and the phrase numbers in which the features appeared.

## 4. Discussion

In this study, we created two groups using clustering HAM-D and separated these groups with 79% accuracy by analyzing the acoustic features of the speech using decision trees.

Because a previous study [21] suggested that the detection of depression may be due to changes in brain activity, such as a decrease in working memory activity, we predicted that groups with different symptoms could be separated using speech. Thus, the ability to separate subjects with different symptoms by speech is the novelty of this study.

Group 1 had significantly higher scores for “guilt”, “work and activities”, and “loss of appetite” than Group 2, as shown in Figure 1. Thus, Group 1 had an inactivity tendency. Meanwhile, “insomnia early” was significantly higher in Group 2. Thus, Group 2 tended toward insomnia. Table 8 shows the features included in the prediction model and the phrase numbers in which the features appeared. The features selected for this model were mostly formant and MFCC, commonly used in speech recognition. As the inactivity group was assumed to be less physically active than the insomnia group, we assumed that features related to motor function were selected.

We separated the two groups with an accuracy of 79%. This was almost comparable to the results of Chlasta et al. [22], who estimated depression using CNN with an accuracy of 77%. They used DAIC, a database published by the University of Southern California. They used data from interviews lasting from 7 to 33 min per session, which were cut into 15 s data. On the other hand, our study used 1 min fixed phrases, and it is conceivable that our data were more homogeneous in content than their data. Although the results were comparable, there were these differences between ours and the previous study.

The data collected in this study were limited to two groups because of the numbers in each group. More data were needed because the symptom groups we created had few symptoms and a low severity. Moreover, symptoms that changed with the course of treatment also needed to be verified. As the data were collected at only one hospital and the patients were relatively elderly, we will collect data on younger patients in the future.

We think this technique can support physicians in their medication decisions. When treating depression, in addition to the basic treatment for depression, such as prescribing antidepressants, it is necessary to alleviate symptoms that patients complain about, for example by prescribing sleeping pills. If the severity of inactivity and insomnia symptoms can be estimated using this technique, physicians can make appropriate medication adjustments based on the presented information.

This technique also visualizes the symptoms of depression in a region and enables the investigation of region-specific factors. Here, visualization means expressing the presence of a symptom as a numerical value. For example, if a region has a high incidence of depression due to insomnia symptoms, one can suspect that factors, such as noise, disturb people’s sleep in that region. Although such a survey can be conducted through questionnaire, we believe that we can find a problem more quickly if we can infer from voice or daily conversations.

The results of this study indicate the potential of this technology for therapeutic and public health applications.

## 5. Conclusions

This study aimed to classify symptoms from voices rather than depression severity and to examine the possibility of separating two groups with a similar depression severity but different symptoms using voices. We separated the two groups with an accuracy of 79% by combining acoustic features and learning with the decision tree. That suggests that it is possible to detect differences in symptoms.

However, the amount of data are insufficient to establish the technique. In the future, we would like to collect data the needed for each symptom as well as data from various medical institutions.

## Figures and Tables

**Figure 1 ijerph-20-03965-f001:**
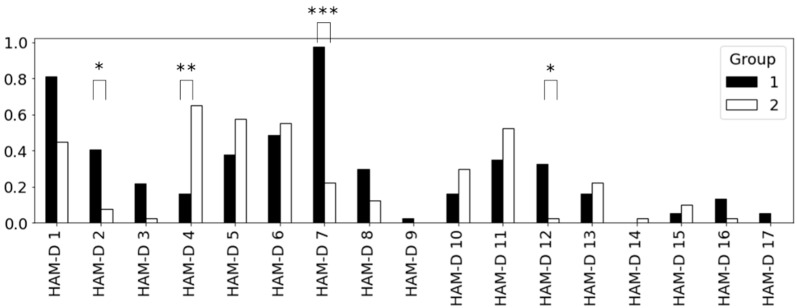
Average score of each item of HAM-D by depression group. There are significant differences in HAM-D 1, 4, 7, and 12.

**Figure 2 ijerph-20-03965-f002:**
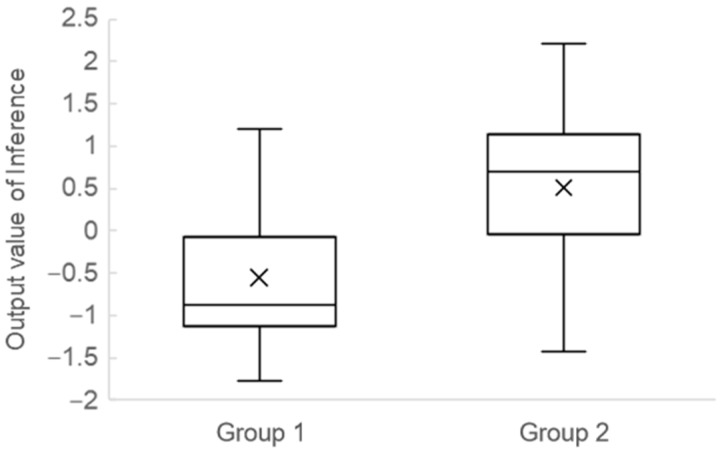
Distribution of the predictive score of the decision tree.

**Table 1 ijerph-20-03965-t001:** Fixed phrases list.

No.	Fixed Phrase (Japanese)	Intention
1	I ro ha ni ho he to	Text that is familiar and easy to pronounce without emotion.
2	Honjitu wa seiten nari	
3	Tsurezurenaru mamani	
4	Wagahai wa nekodearu	
5	Mukashi mukashi arutokoroni	
6	Omoeba tookue kitamonda	
7	A i u e o	Articulation test of vowel
8	Ga gi gu ge go	Articulation test of consonant [g]
9	La li lu le lo	Articulation test of consonant [l]
10	Pa pi pu pe po	Articulation test of consonant [*p*]
11	Galapagos shoto	Word containing various consonants
12	Tsukarete guttari shiteimasu	Emotional symptoms in DSM 5
13	Totemo genkidesu	
14	Kinouha yoku nemuremashita	
15	Shokuyoku ga arimasu	
16	Okorippoi desu	
17	Kokoro ga odayakadesu	
18	Ah (sustained over 3 s.)	Sustained vowel
19	Eh (sustained over 3 s.)	
20	Uh (sustained over 3 s.)	
21	Pa-ta-ka… (for 5 s as quickly as possible)	Oral diadochokinesis

**Table 2 ijerph-20-03965-t002:** Items of GRID-HAM-D_17_.

No.	Item
1	Depressed mood
2	Guilt
3	Suicide
4	Insomnia early
5	Insomnia middle
6	Insomnia late
7	Work and activities
8	Psychomotor retardation
9	Psychomotor agitation
10	Anxiety, psychic
11	Anxiety, somatic
12	Loss of appetite
13	Somatic symptoms, general
14	Sexual interest
15	Hypochondriasis
16	Loss of weight
17	Insight

**Table 3 ijerph-20-03965-t003:** Acoustic features for the prediction and libraries for the calculation.

Type	Feature Name	Library
Glottal flow	NAQ	DisVoice
	QOQ	
Frequency	FFT Band Power	openSMILE
	FFT Spectral Centroid	
	FFT Spectral Flux	
	Formant	
	MFCC	
Prosody	Energy RMS	openSMILE
	Fundamental Frequency	praat
	HNR	
	Jitter	
	Shimmer	

**Table 4 ijerph-20-03965-t004:** Number of subjects according to generation.

Age	Male	Female
20–29	1	1
30–39	5	6
40–49	7	12
50–59	15	11
60–69	8	13
70–79	6	21
80–89	1	3
total	43	67

**Table 5 ijerph-20-03965-t005:** The results of the clustering of subjects from group 1. The number and the average age of each group. This table shows no significant differences between these profiles.

	Number			Age		
Group	All	Male	Female	All	Male	Female
1	37	11	26	54.8 ± 13.7	48.7 ± 11.5	57.4 ± 13.9
2	40	18	22	57.6 ± 14.9	52.2 ± 14.5	62 ± 14.1

**Table 6 ijerph-20-03965-t006:** The average of HAM-D and medication dose. The medication dose values are converted to the equivalent of antidepressant imipramine. These properties also show no significant difference.

	HAM-D	Medication Dose (mg)
Group	Average	Average
1	5	70.4
2	3.9	90.6

**Table 7 ijerph-20-03965-t007:** Accuracy of decision trees.

*n* = 77	Predicted Group 1	Predicted Group 2
Actual Group 1	31	6
Actual Group 2	10	30

**Table 8 ijerph-20-03965-t008:** Common acoustic features in the decision trees.

Feature Name	Statistics	Fixed Phrase No.
Formant 1st	mean	7
Formant 2nd	std *	4
Formant 2nd	mean	21
Formant 4th	mean	6, 21
MFCC 1st	mean	14
MFCC 4th	std	14
MFCC 6th	mean	4, 15, 17, 20
MFCC 7th	std	11
MFCC 8th	std	18
MFCC 8th	mean	11, 19
MFCC 10th	std	8
MFCC 12th	mean	5

* std: standard deviation.

## Data Availability

The data presented in this study are available upon request from the corresponding author. The data are not publicly available due to privacy and ethical restrictions.

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
