# Peer review of "Estimating Depressive Symptom Class from Voice"

_ijerph, 2023, doi:10.3390/ijerph20053965_

Round 1

Reviewer 1 Report

Thank you for the opportunity to review this intriguing paper on voice biomarkers in depression. This research is innovative, with potential practical implications but several aspects of the research need clarifications and corrections in the paper so I kindly ask the authors to address them all.

  • The Abstract should be structured to 1) Background, 2) Methods, Results, and 4) Conclusion. No results are stated in the Abstract.
  • Page 1. Line 36. “can diagnose depression”. Diagnosis of depression is made based on diagnostic criteria, such as DSM-5, and tests should be used to detect the presence and grade severity of symptoms and signs of depression. Please rephrase this statement to acknowledge that the diagnosis can not be made solely by the use of listed tests and that it is not the tests that make the diagnosis but a trained physician according to defined and accepted clinical criteria.
  • Page 2. Line 61 “Scherer et al.” appears twice, please correct
  • Please provide more information on patient selection for the study. How did you select patients? What was the setting, hospitalized patients or those in daily clinics or outpatient clinics? Did you include all consecutive patients with depression and in what period? Were patients on medication at the time of voice evaluation, and if so on what types of medication?
  • Who performed voice analysis and in what setting? Were the operator blinded for the diagnosis of depression? What was the duration of the voice recording and was it done in one session?
  • Why didn’t you include a control group of subjects without depression?
  • Please provide a brief explanation of the features of the acoustic analysis.
  • Table 1. Demographic data on subjects (sex, age) should be included in the Results section of the paper, not in the Methods. Information from Table 1 is repeated later so this table should be deleted and information on the sex and age of the participants should be mentioned in the text at the begging of the Results section of the manuscript.
  • Table 4 should be in the Results section of the manuscript. 

  • The Results section of the manuscript should start with basic information regarding your dataset as mentioned previously.
  • Please define abbreviations FFT, MFCC, RMS, etc. in Table 3 and the text. Please double-check all abbreviations in the text.
  • Table 7 is redundant, this information can be stated in the text of the manuscript.
  • Page 5, lines 130,131, “were selected from Table 4”, did you mean from Table 3?
  • The discussion section of the paper needs to be expanded. Page 7, line 188, the first sentence in the paragraph “Next, we discuss the prediction…” is redundant and should be deleted. Please compare your results with previous work and discuss the differences. Please state explicitly the advantages and novelty as well as limitations of your research compared to published literature. The Conclusion section should be shortened, it should be brief and focused only on your main findings. Some statements from the Conclusion should be moved to the Discussion.  

Reviewer 2 Report

The paper is well written and define a very interesting way to estimate depressive symptoms.

In this work the authors classified symptom classes based on HAM-D scores and estimated patients’ classes from voice.

Comments:

1.     At row 29 you reported some statistics of depression, but you didn’t mentioned any sites or papers. Can you insert some reference that support statistics?

2.     At row 38 you wrote that the subjectivity of physicians and patients may influence the diagnosis. Why? Can you provide some example?

3.     From  row 57 you described the IAIF method, but it’s not clear its impact for the scope. Can you explain in detail?

4.     In the 2.2.1 chapter you described the subjects involved in this study and you specified the percentage of man and woman and the age average. It can be useful to understand the people affected by depression in this pilot study also specifying the nationality and grouping people for age (using some ranges for instance: 18-35 years old, 36-50, 51-70, 71-….).

5.     At row 125 you wrote that you used K-means for classify the unlabeled data. K-means is not a Classification Method, but only clustering. How do you use K-means for classification? Did you group people using clustering and then assign to each cluster a class? If you use this method, I suggest you cite some work already done that combine clustering and classification techniques:  

Comparative Analysis of Clustering Algorithms and Moodle Plugin for Creation of Student Heterogeneous Groups in Online University Courses doi :  https://doi.org/10.3390/app11135800

6.     At row 130 you mentioned “previous studies” but you didn’t insert any reference. Can you cite some paper that described studies related the acoustic features used for training?

7.     The label caption of the table 4 is very short. I suggest to expand this part commenting in details the results, in order to be more understandable for the reader.

Round 2

Reviewer 1 Report

  1. The first sentence of the Abstract is not necessary for the study and should be deleted. The Abstract should be corrected, and only the most essential information should be left.
  2. "Jitter, and Shimmer,..." Did you mean that the research or analysis of Jitter and Shimmer led to the finding "that Jitter was more significant..." Please correct.
  3. "After Alku published...." this sentence should be rephrased, e.g., "After... it is possible to estimate glottal flow" or "...glottal flow estimation is possible".
  4. "We recruited patients diagnosed with depression..." - please state details; what was the diagnosis? Was it major depression?
  5. "They invited..." Who invited patients? Please rephrase. 
  6. Please explicitly state the inclusion and exclusion criteria for your study.
  7. In the Discussion section, you mention for the first time "inactivity group as group 1—groups are not defined clearly. How did you create groups, and according to this methodology, which patients went to Group 1 and which to Group 2? Please explain the clustering of the HAM-D scores. 
  8. This Graph is not numbered and still contains the term "class" instead of "group".
  9. The first sentence of this paragraph (the fifth paragraph in the Discussion, marked in yellow) should be the second sentence of the Discussion part of the manuscript.
  10. The marked sentence in the sixth paragraph of the Discussion is not clear to me, and you did not study two diseases but one, depression.
  11. What do you mean by visualizing symptoms of depression? Please explain.
  12. The Discussion section of the paper should still be improved and expanded with the Discussion on the clinical implications of your findings, as they may assist clinicians in supporting the diagnosis of depression or detecting depressive features and cluster patients in subgroups. 
  13. English editing is needed. 
    Please see all comments marked in the attached file. The authors substantially improved the manuscript, but these additional issues should be addressed. Thank you.
